



# GEMS-GER: A Machine Learning Benchmark Dataset of Long-Term Groundwater Levels in Germany with Meteorological Forcings and Site-Specific Environmental Features

Marc Ohmer[1], Tanja Liesch[1], Bastian Habbel[1], Benedikt Heudorfer[2], Mariana Gomez[3], Patrick Clos[3], Maximilian Nölscher[3], and Stefan Broda[3]

[1]Institute for Applied Geosciences (AGW), Karlsruhe Institute of Technology (KIT), Karlsruhe, Germany
[2]Institute of Meteorology and Climate Research - Atmospheric Trace Gases and Remote Sensing (IMKASF), Karlsruhe Institute of Technology (KIT), Karlsruhe, Germany
[3]Federal Institute for Geosciences and Natural Resources (BGR), Berlin, Germany

**Correspondence:** Marc Ohmer (marc.ohmer@kit.edu)

**Abstract.** We present GEMS-GER (Groundwater Levels, Environment, Meteorology, Site Properties), the first benchmark dataset specifically designed for machine learning applications in long-term groundwater level modeling in Germany. The dataset comprises 32 years of gapless weekly observations from 3,207 monitoring wells, enriched with meteorological forcing variables and more than 50 site-specific static attributes. All data have undergone extensive preprocessing, including harmonization, outlier removal, and iterative imputation, to ensure high quality and suitability for machine learning applications. The wells are spatially distributed across Germany and cover diverse hydrogeological settings and aquifer types. To demonstrate the utility of the dataset, we provide three initial benchmark models: a single-well CNN model, a global LSTM model using dynamic inputs, and a global LSTM model incorporating both dynamic and static features. The best-performing model achieves satisfactory predictive performance (NSE > 0.5) for more than half (52%) of the wells, which is considered a strong result in the context of groundwater modeling.

GEMS-GER is openly available under an open-access license via Zenodo, accompanied by detailed documentation. By enabling standardized and reproducible evaluation of data-driven groundwater models, the dataset offers a robust foundation for advancing machine learning research in hydrogeology.

## 1 Background and Motivation

Groundwater is a vital resource in the global supply of drinking water, agriculture, and ecosystems. In Germany, groundwater, including water from springs, accounts for approximately 70% of the drinking water supply (Destatis, 2025). However, unlike surface water, it is a hidden resource that cannot be directly observed, with data collection primarily limited to discrete measurements from wells and springs. This spatial and temporal discontinuity often hinders understanding system-wide processes and the ability to respond to climatic or anthropogenic influences. Reliable forecasts enable decision-makers in policy and management to respond proactively to potential risks such as water scarcity, over-extraction, or contamination (De Graaf et al., 2019). Moreover, accurate groundwater level (GWL) predictions facilitate more effective water resource management by



balancing ecological requirements with urban growth and industrial development demands (Wunsch et al., 2021; Gomez et al., 2024). Finally, long-term strategies for mitigating the impacts of climate change, such as shifts in precipitation patterns and rising temperatures, depend heavily on robust GWL projections, ensuring sustainable socio-economic development (Destatis, 2025; Shaikh and Birajdar, 2024).

Despite its importance, groundwater level forecasting remains challenging due to the groundwater systems' hidden, interconnected nature, which is influenced by various physical, geological, and climatic factors (Ahmadi et al., 2022; Feng et al., 2024). While physics-based numerical models can represent these processes in considerable detail, they are associated with significant demands. They require extensive and often costly input data, including physical properties that are inherently uncertain. Their setup is complex, demanding specialized expertise for tasks such as numerical discretization and the definition of boundary and initial conditions, along with time-intensive calibration and validation phases (Chen et al., 2020).

Machine Learning (ML) has proven to be a highly effective approach in hydrological modeling, particularly for groundwater level prediction. ML addresses several limitations of physics-based methods by capturing complex non-linear relationships between hydro-climatic variables, even when observational data are limited. Early studies, such as Coulibaly et al. (2001) and Lallahem et al. (2005), compared different Artificial Neural Networks (ANNs) architectures, demonstrating their capability to simulate monthly GWL using climatic and hydrological data. In the following years, interest grew in various ANN models, including feedforward neural networks (FFNNs) like multilayer perceptrons (MLPs) (Nayak et al., 2006; Krishna et al., 2008) and radial basis function networks (RBFNNs) (Ying et al., 2014; Chen et al., 2010), for GWL prediction across diverse hydrogeological settings. According to a review by Tao et al. (2022), the number of publications on AI methods in GWL modeling has significantly increased, particularly since the mid-2000s, reflecting the growing recognition of ML's ability to capture the complex, non-linear patterns in GWL fluctuations. The field has evolved to incorporate more advanced ML techniques. For example, Adaptive Neuro-Fuzzy Inference Systems (ANFIS) (Kholghi and Hosseini, 2009; Emamgholizadeh et al., 2014; Saumen and Tiwari, 2014) have gained popularity by combining neural learning with fuzzy logic. Similarly, support vector machines (SVMs) and support vector regression (SVR) have frequently been used for GWL forecasting (Huang et al., 2017; Guzman et al., 2019; Yoon et al., 2011). Another emerging trend is the development of hybrid models that integrate various techniques to harness their strengths and mitigate their weaknesses (Tao et al., 2022). For example, wavelet-based hybrid models (Moosavi et al., 2013; Samani et al., 2022; Barzegar et al., 2017) combine wavelet analysis with AI algorithms. Optimization-enhanced models employ metaheuristic algorithms, such as Genetic Algorithms (Kasiviswanathan et al., 2016; Sadat-Noori et al., 2020), to refine model parameters and architectures. Recently, deep learning architectures such as convolutional neural networks (CNNs) and long short-term memory networks (LSTMs) (Wunsch et al., 2021, 2022a; Heudorfer et al., 2024; Han et al., 2025; Solgi et al., 2021; Yang and Zhang, 2022) have attracted increasing attention for their ability to capture complex temporal dependencies and long-term trends in GWL data.

Despite significant methodological progress, the field still lacks standardized, large-scale benchmark datasets. Most studies rely on localized data, which often fail to capture the diversity of hydrogeological and climatic conditions, limiting the transferability of results. In addition, many datasets are not publicly available or lack proper documentation, impeding reproducibility





and validation. These limitations hinder systematic comparison and generalization of existing models, highlighting the need to further explore their application across diverse hydrogeological settings.

In contrast, using comprehensive, standardized, and multivariable datasets with long-term observations has become a well-established practice in other areas of hydrology. The CAMELS (Catchment Attributes and Meteorology for Large-Sample Studies) dataset series is a notable example. CAMELS was initially introduced with CAMELS-US (Addor et al., 2017) for the United States, integrating hydrological, meteorological, and catchment-specific characteristics. Over time, additional national and multinational variants have been developed to address region-specific needs. On a global scale, the Caravan dataset (Kratzert et al., 2023) unites regional CAMELS datasets into a standardized format, facilitating global hydrological modeling and cross-regional comparisons.

However, to our knowledge, no comparable dataset optimized for machine learning–based groundwater modeling currently exists. To fill this gap, this study introduces a machine learning–ready, comprehensive, large-scale, and nationally standardized dataset for groundwater level modeling and prediction in Germany. It contains 32 years of gapless weekly groundwater level time series from over 3,000 monitoring wells, derived through preprocessing from more than 17,000 original wells. These time series are complemented by meteorological forcing data and over 50 site-specific static features per well, covering hydrogeological and hydrological properties, soil characteristics, land use, topography, and monitoring well metadata.

The dataset is designed to optimally support the application of machine learning models while meeting high standards for comparability, reproducibility, and generalizability. It follows the established principles and structure of the CAMELS datasets, adapting them to the requirements of groundwater modeling. The overarching objective of this work is to create a central data foundation that enables systematic comparison of machine learning models, validates their generalizability across various hydrogeological and climatic conditions, and ensures transparent research through public availability and standardized documentation. The provision of this dataset represents a crucial contribution to advancing data-driven research in groundwater modeling.

In addition to the dataset, we present the results of three benchmark model types: (1) individual single-well models, each trained separately for a specific monitoring well using only dynamic inputs; (2) a global model trained on all wells, also using dynamic inputs; and (3) a second global model that incorporates both dynamic and static inputs.

The core objectives of the dataset and benchmark models are:

- Establish a large-scale comprehensive dataset for groundwater modeling, integrating 32 years of weekly groundwater level data from Germany and meteorological forcings and site-specific properties.

- Enable reproducible evaluation of machine learning models for groundwater level prediction through a dataset for direct performance comparisons, and give some first benchmark model results.

- Bridge local and global groundwater research by providing a unified dataset for model development and validation across spatial scales.

- Promote transparency and collaboration via public access, thorough documentation, and standardized data formats.



## 2 Data and Preprocessing

### 2.1 Data sources

The groundwater level data were obtained from the responsible authorities of the 16 German federal states, namely: Landesanstalt für Umwelt Baden-Württemberg (LUBW), Bayerisches Landesamt für Umwelt (LfU), Senatsverwaltung für Mobilität, Verkehr, Klimaschutz und Umwelt Berlin (SenMVKU), Landesamt für Umwelt Brandenburg (LfU), Geologischer Dienst für Bremen (GDfB), Behörde für Umwelt, Klima, Energie und Agrarwirtschaft Hamburg (BUKEA), Hessisches Landesamt für Naturschutz, Umwelt und Geologie (HLNUG), Landesamt für Umwelt, Naturschutz und Geologie Mecklenburg-Vorpommern (LUNG), Niedersächsischer Landesbetrieb für Wasserwirtschaft, Küsten- und Naturschutz (NLWKN), Landesamt für Natur, Umwelt und Klimaschutz Nordrhein-Westfalen (LANUV), Landesamt für Umwelt Rheinland-Pfalz (LfU), Landesamt für Umwelt- und Arbeitsschutz Saarland (LUA), Sächsisches Landesamt für Umwelt, Landwirtschaft und Geologie (LfULG), Landesbetrieb für Hochwasserschutz und Wasserwirtschaft Sachsen-Anhalt (LHW), Landesamt für Umwelt Schleswig-Holstein (LfU), and Thüringer Landesamt für Umwelt, Bergbau und Naturschutz (TLUBN). The original dataset comprised groundwater level time series from more than 17,000 monitoring wells.

The meteorological data originate from two main sources. Variables from the HYRAS dataset provided by the German Meteorological Service (DWD) include mean, maximum, and minimum daily temperature (DWD, 2024a, b, c), daily precipitation sum (DWD, 2024d), and relative humidity (DWD, 2024e), as well as real, potential, and reference (FAO) evapotranspiration (DWD-CDC, 2024a, b, c), soil moisture (DWD-CDC, 2024d), and soil temperature at 5 cm depth (DWD-CDC, 2024e). Additional variables (snow water equivalent, snowfall, and snowmelt) were obtained from ERA5-Land data (Muñoz-Sabater et al., 2021), which also include hydrological fluxes such as surface and subsurface runoff. The latter are simulated from atmospheric forcing using a land surface model. Table 1 provides an overview of all included dynamic features and their data sources.

The site-specific static data include well metadata such as coordinates, well depth, screen length (if available), aquifer type, and pressure condition, which were also obtained from the responsible authorities. For wells located in North Rhine-Westphalia (LANUV), spatial coordinates were anonymized by rounding to a horizontal resolution of 100,/m in accordance with applicable data protection regulations. In addition, we incorporated static features capturing hydrogeological and soil characteristics (e.g., aquifer type, hydraulic conductivity, soil type, recharge), topographic attributes (elevation, slope, aspect, flow direction), and land use information. An overview of all static features and their corresponding data sources is provided in Table 2. Mean climatic variables were not included, as they can be easily derived from the dynamic inputs through feature engineering.

### 2.2 Groundwater level data

#### 2.2.1 Data Integration and Weekly Aggregation

As illustrated in Figure 1a, the data preparation process involved harmonizing datasets of varying formats and structures originating from the different data management systems of the 16 federal states into a unified format. Subsequently, all time series were aggregated to weekly means where higher temporal resolutions were available. For consistency, the weekly aggregation





was aligned to Mondays across all datasets. In cases of individual measurements within a week, values were also assigned to the corresponding Monday to ensure temporal alignment.

### 2.2.2 Data Gap Filtering

Filtering criteria (Fig. 1b) were defined to balance data quality with spatial and temporal coverage:

– **Time Period:** The period from 1991 to 2022 was selected to capture long-term trends while ensuring data recency. More recent data were incomplete in many time series.

     – **Missing Values:** Monitoring wells with more than 20% missing data were excluded.

     – **Maximum Gap Length:** Wells with continuous data gaps exceeding 12 weeks were excluded to preserve the integrity of the time series.

### 2.2.3 Sudden Change Detection

Abrupt shifts in groundwater level time series (Fig. 1c) were identified using the PELT (Pruned Exact Linear Time) algorithm (Killick et al., 2012), implemented via the `Ruptures` library (Truong et al., 2020). This method partitions time series into segments of consistent statistical properties by minimizing intra-segment variance through a cost function. The analysis focused on significant step-like level changes over the 32-year period. All detected changepoints were manually reviewed to assess their plausibility and potential origin. Only those shifts that could not be explained by natural groundwater fluctuations — and were likely attributable to anthropogenic influences such as data logger repositioning, construction activity, or technical malfunction — were considered grounds for excluding the affected time series from further analysis.

### 2.2.4 Multi-Criteria Outlier Detection

To ensure reliable identification of implausible values, a multi-criteria outlier detection approach (Fig. 1d) was implemented. 140 Five distinct algorithms were applied in parallel to leverage their individual strengths and compensate for their weaknesses. This ensemble strategy reflects a conservative approach: only data points consistently identified as anomalous were considered for removal, minimizing the risk of excluding valid measurements. A value was flagged as a potential outlier only if at least four out of five methods classified it as implausible. These flagged data points were then subjected to manual visual inspection to assess their plausibility. Only those confirmed as clearly erroneous were ultimately removed. The following detection methods 145 were employed:

1. **Isolation Forest:** A tree-based method that evaluates how easily a data point can be isolated from the rest of the dataset, particularly effective for identifying global anomalies (Liu et al., 2008, 2012).

2. **Local Outlier Factor (LOF):** Identifies local outliers by comparing the density of a data point to that of its neighbors (Breunig et al., 2000).





3. **Seasonal Decomposition:** Decomposes time series into trend, seasonal, and residual components. Anomalies were defined as residuals exceeding four standard deviations from the mean (Seabold and Perktold, 2010).

4. **Long-term Z-Score Analysis:** Based on a 26-week moving average, identifying points with Z-scores greater than 3 as potential outliers.

5. **Short-term Z-Score Analysis:** Uses an 11-week moving average to detect short-term deviations. Z-scores above 2.65 were flagged as outliers.

### 2.2.5 Data Imputation

Data gaps of a maximum of 12 weeks were imputed using the *Iterative Imputer* (Buuren and Groothuis-Oudshoorn, 2011; Buck, 1960) from the `scikit-learn` library (Pedregosa et al., 2011a), implementing a multivariate imputation strategy based on the relationships between correlated monitoring wells (Fig. 1e). The imputation relied on a *Bayesian Ridge* estimator, which incorporates uncertainty in the parameter estimates and has been shown to perform well in the presence of multicollinearity (MacKay, 1992; Tipping, 2001). To account for temporal variability and to improve imputation accuracy, the dataset was partitioned into six overlapping blocks, each covering approximately six years, with a three-month temporal overlap. Imputation was performed independently for each block. Monitoring wells that were excluded from the primary analysis (Fig. 1b) due to extended data gaps were conditionally reintroduced as auxiliary predictors, provided they exhibited substantial correlation with other wells during the respective time block. To ensure a minimum level of reliability, only those auxiliary wells with less than 25% missing values within the given block were included. Although not included in subsequent analyses, these auxiliary wells were assumed to provide additional contextual information that could potentially support more accurate estimation of missing values in the target wells. For each target monitoring well to be imputed, the 200 most highly correlated wells, based on overlapping time periods, were selected as predictors. Imputation was performed in successive six-year time blocks with a temporal overlap of three months between adjacent blocks. The six-year window balances the need to capture seasonal patterns with ensuring sufficient data availability in auxiliary wells. Longer periods would increase the risk of excluding wells due to missing data. Overlapping imputations were averaged, and blocks were merged into a continuous dataset.

### 2.2.6 Results of Preprocessing and Dynamic Groundwater Time Series Analysis

Figure 2 illustrates the evolution of the dataset through the processing steps described in 2.2.1 (Data Integration and Weekly Aggregation), 2.2.2 (Data Gap Filtering), and 2.2.5 (Data Imputation). The heatmaps illustrate the number of groundwater monitoring wells per German federal state throughout the preprocessing steps: before filtering, after filtering, and after imputation. Due to extensive data gaps and the absence of sufficiently long time series, no monitoring wells from Bremen (HB), Hamburg (HH), or Saarland (SL) were included in the final dataset. On average, 1.05% of the values were imputed, with imputation rates per well ranging from 0% to 1.2%.

In total, 3,207 weekly groundwater level time series were retained and enriched with dynamic indicators. During Data Gap Filtering (Section 2.2.2), 10,842 wells were excluded due to insufficient data coverage. Sudden Change Detection (Sec-





tion 2.2.3) led to the removal of 32 implausible wells. Multi-Criteria Outlier Detection (Section 2.2.4) flagged 379 potential outliers, of which 57 individual observations were discarded following visual plausibility checks.

Figure 3 provides spatial context for assessing regional groundwater dynamics. Panels 3a–b show the ten Major Hydroge-
ological Districts (MHDs), which encompass a range of aquifer types, from porous, unconsolidated deposits in the northern lowlands to fractured and karstified bedrock systems in upland regions. The highest monitoring well density occurs in the Northern and Central German Unconsolidated Rock District (MHD1) and the Upper Rhine Graben with the Mainz Basin (MHD3), both of which are characterized by thick sedimentary sequences forming highly productive aquifers that are critical for regional water supply.

To characterize groundwater dynamics across these diverse settings, a set of time series–based indicators was computed. The indicators shown in Panels 3c–i are described in detail by Wunsch et al. (2022b) and Richter et al. (1996), and quantify key aspects of groundwater variability. *SD_diff* captures short-term fluctuations via the standard deviation of first-order dif-ferences, reflecting the volatility of daily to weekly changes. *range_ratio* is the ratio of mean annual to total range, indicating the proportion of overall variability explained by interannual fluctuations. *ex_vals* denotes the relative frequency of identified
peaks, serving as a proxy for abrupt, high-magnitude events such as recharge pulses. *seasonal_behaviour* quantifies the simi-larity between the monthly mean cycle and a sinusoidal annual curve, measuring the strength of seasonal dynamics. *periodicity* captures intra-annual regularity by correlating the series with its weekly climatology. *yearly_variance* reflects the median of annual variances, providing a robust estimate of typical seasonal amplitude. Lastly, *HPD* (High Pulse Duration) measures the cumulative duration of groundwater levels above the long-term mean, indicating the persistence of high-water phases often
linked to extended recharge periods.

Figure 4 shows an overview of selected dataset variables, including spatial representations of both dynamic (aggregated as long-term means) and static features. The dynamic layers include mean groundwater level, mean annual precipitation, and potential evapotranspiration for the period 1991–2022, while the static layers illustrate key site characteristics such as soil group, porosity type, hydrogeological region, organic matter content, land use, and groundwater recharge. These visualizations
highlight the spatial heterogeneity of input variables and emphasize the multivariate nature of the dataset.

## 2.3 Meteorological forcing data

Meteorological data were obtained from the respective providers and extracted as point values at the exact locations of the remaining groundwater monitoring wells. All time series were temporally aligned and resampled to a consistent weekly res-olution matching that of the groundwater data. Depending on the variable, either weekly means or sums were applied during
aggregation. Details on the variables used, including their data sources and aggregation types, are summarized in Table 1.

## 2.4 Site-specific static data

All spatial datasets were first harmonized to a common coordinate reference system (CRS), with EPSG:3035 – ETRS89-extended / LAEA Europe. Selected raster layers underwent additional preprocessing, including the generalization of the digital elevation model from 1 m resolution (DTM1) to a coarser 20 m resolution (DTM20). This resampling step was performed

to reduce high-frequency noise, improve the numerical stability of terrain derivatives, and decrease computational demand, particularly for hydrologically relevant terrain metrics derived using the SAGA GIS framework (e.g., slope, aspect, curvature, flow direction, and flow accumulation). The 20 m resolution was considered a suitable compromise between preserving relevant topographic detail and achieving robust parameterization at the landscape scale. In cases where individual raster layers contained small data gaps (i.e., isolated no-data cells), these were interpolated using the `gdal_fillnodata` utility with in-
verse distance weighting (IDW). Finally, all spatial variables were extracted at the geographic coordinates of the groundwater monitoring wells for subsequent integration with time series data.

## 3    Dataset Structure

The **GEMS-GER** (Groundwater, Environment, Meteorology, Site-properties – Germany) dataset is structured into two primary components: dynamic time series and static site descriptors. In addition, it includes benchmark model outputs to support
reproducible model evaluation.

### 3.1    Dynamic time series data

The dynamic data, stored in the `GEMS-GER_data/dynamic/` directory, consist of individual files for each of the 3,207 monitoring wells, named using the pattern `MW_{ID}.csv`. Each file contains weekly aggregated groundwater level observations (GWL) for Mondays from 1991 to 2022.
Alongside GWL, the files include a wide range of meteorological and hydrological forcing variables as summarized in Table 1. These include daily mean, maximum, and minimum temperature, precipitation, and relative humidity from the HYRAS dataset provided by the German Meteorological Service (DWD), as well as real, potential, and reference (FAO) evapotranspiration, soil moisture, and soil temperature at 5 m depth. Further variables such as snow water equivalent, snowfall, snowmelt, and surface and subsurface runoff are derived from the ERA5-Land dataset. A binary column, `GWL_flag`, indicates whether
a GWL value was directly observed (`True`) or imputed (`False`).

An example plot of selected dynamic timeseries (groundwater level, precipitation, temperature, evapotranspiration and runoff) is shown in the Appendix for well `MW_1` (Figure A1). Corresponding illustrations for all wells are included in the dataset in the `GEMS-GER_figures/` directory.

### 3.2    Static site descriptors

The static data, located in the `GEMS-GER_data/static/` directory, are provided in a single file, `static_features_MW_1toMW_3207.csv`, containing temporally invariant attributes for each monitoring well. These include hydrogeological, hydrological, soil, land use, and geomorphological descriptors, as listed in Table 2.





### 3.3 Benchmark model performance

Model performance metrics for the three benchmark models introduced in Section 4 are stored in the `GEMS-GER_data/`

`model_performance/` directory. Each file contains the median values of four standard metrics (NSE, RMSE, $R^2$, and Bias) across ten model runs:

- `model_performance_single.csv` — Single-well CNN models

- `model_performance_global_dynonly.csv` — Global LSTM (dynamic inputs only)

- `model_performance_global_dynstat.csv` — Global LSTM (dynamic + static inputs)

## 4 Benchmark models

We implemented three types of benchmark models: (i) single-well models for each monitoring well, using dynamic inputs only, (ii) a global model (i.e., one model for all monitoring wells, also referred to as a regional model) using dynamic inputs only, and (iii) a global model using both dynamic and static inputs.

The single-well models are based on a Convolutional Neural Network (CNN) architecture, which has previously shown

good performance in modeling groundwater level time series (Wunsch et al., 2021, 2022a; Gomez et al., 2024). The global models are largely based on the Long Short-Term Memory (LSTM) architecture used in Heudorfer et al. (2024), with minor modifications. The model using only dynamic inputs is a straightforward LSTM, while the model incorporating both dynamic and static inputs consists of two branches: an LSTM branch for dynamic inputs and a Multi-Layer Perceptron (MLP) branch for static inputs, which are concatenated prior to the output layer.

We deliberately refrained from hyperparameter optimization and employed relatively simple, yet established and proven architectures. The goal was not to achieve optimal prediction performance, but to provide a robust and transparent benchmark for future modeling studies.

These models also help identify monitoring wells where performance based solely on dynamic meteorological inputs is insufficient. This may indicate the relevance of other dynamic drivers (e.g., groundwater abstraction, surface water interactions),

or specific hydrogeological conditions, such as thick unsaturated zones or deep aquifers with low-permeability confining layers. In such cases, the 52-week input window may be too short to capture relevant dynamics.

Furthermore, comparing the three benchmark models enables an assessment of how model performance improves through the integration of additional data in the global model compared to single-well models, and the specific contribution of static features.

All models were evaluated on the last 10 years of the time period (2013–2022). The remaining data were used for training (1991–2007) and validation with early stopping (2008–2012). The input sequence length of the dynamic inputs is 52 weeks (i.e., one year) for all models. All metrics were computed on the median prediction of an ensemble of ten model initializations.





### 4.1 Model Setup

#### 4.1.1 Single-well models

The single-well models consist of a Convolutional Neural Network (CNN) with the following architecture: one hidden Convolutional layer with 256 filters and a kernel size of 3, followed by a MaxPooling layer, a Flatten layer, a Dense layer with 32 units, and a final Dense output layer with a single unit. The models are trained using the Adam optimizer with a learning rate of 0.001. Training is performed for a maximum of 30 epochs, with early stopping (patience of 5 epochs). A batch size of 16 is used. All available dynamic input features are provided to the models.

#### 4.1.2 Global models

Both global models use all available dynamic input features. The model architecture consists of a single Long Short-Term Memory (LSTM) layer with 128 units and a dropout rate of 0.3. Training is conducted using a batch size of 512 for a maximum of 20 epochs, with early stopping (patience of 5 epochs). A learning rate scheduler is applied, targeting a final learning rate of 0.001. The global model that also incorporates static input features includes a second model branch in addition to the LSTM

component. This branch processes the static inputs via a Dense layer with 128 units. The outputs of the LSTM and static input branches are concatenated and followed by a Dense layer with 256 units and a final Dense output layer with a single unit. Among the available static features, geographic coordinates, depth, screen information, and pressure state were excluded. These attributes were included in the dataset for completeness but were presumed to be of limited relevance for model performance due to their sparse availability across monitoring wells. Categorical static features were label-encoded.

### 4.2 Model Results

The summarized model results are presented in Table 3 and Figure 5. Detailed performance metrics for each monitoring well and model are available in the file `model_performance.csv` included in the dataset.

The highest median Nash–Sutcliffe Efficiency (NSE) across all wells was achieved by the single-well models, with a value of 0.52, closely followed by the global model with both dynamic and static inputs (median NSE = 0.50). While these values may appear low compared to typical surface water modeling benchmarks, they are considered relatively strong in the context

of groundwater modeling. This is due to the well-documented fact that groundwater level time series exhibit significantly more heterogeneous and complex dynamics than surface water discharge data, making them inherently more difficult to predict.

In all model variants, the mean NSE values are substantially lower than the medians, suggesting that model performance is strongly affected by a subset of poorly performing wells. The highest maximum NSE of 0.94 was also achieved by the

single-well models, followed by the global model that incorporates both dynamic and static inputs (NSE max = 0.91).

A comparison of the two global model variants shows, as expected, that the inclusion of static input features leads to improved performance. The global model with both dynamic and static inputs achieves a mean and median NSE of 0.39 and 0.50, respectively, compared to 0.32 and 0.44 for the model using dynamic inputs only. In terms of the number of wells





with acceptable performance (NSE > 0.5), the model with static inputs performs significantly better, producing 1,583 wells
(approximately 49 %) above this threshold, compared to 1,270 wells (around 40 %) for the model with dynamic inputs only.
The highest number of wells with acceptable performance is again achieved by the single-well models, with 1,669 wells (52 %).
Moreover, the single-well models also yield the largest number of wells with very high predictive performance (NSE > 0.8),
totaling 191 wells. This is followed by the global model with static inputs, which achieves this level of performance for 112
wells.

Wells with negative NSE values (NSE < 0), indicating poor model performance, number 404 for the single-well models (just
under 13 %), 471 for the global model using dynamic inputs only (15 %), and 396 for the global model including static inputs
(approximately 12 %).

    All three model variants share a common subset of 256 wells in this low-performance group, suggesting that the groundwater
dynamics at these sites cannot be adequately captured using the available input features. As discussed previously, potential
reasons for this include anthropogenic influences such as groundwater abstraction, surface sealing, or infiltration from surface
waters, as well as hydrogeological factors like thick unsaturated zones or confined aquifers.

    Figure 6 displays the spatial distribution of NSE values across all wells and model variants, as well as the difference in
performance ($\Delta$NSE) between the single-well model and the global model with static input features.

    At first glance, the spatial patterns of NSE values appear similar across the three model variants, with only minor differences.
In particular, wells with low model performance are distributed comparably. A prominent example is the northern part of the
Upper Rhine Graben (Hessisches Ried), where extensive groundwater management through extraction and infiltration likely
affects model performance. Another example is the Berlin metropolitan area, where dewatering activities influenced several
groundwater level time series during construction projects within the observation period.

    The comparison of the single-well model and the global model with static inputs in terms of $\Delta$NSE reveals that, for the
majority of wells, performance differences are relatively small. Most wells exhibit $\Delta$NSE values within a range of $\pm$0.1–0.2,
indicated by light colors in the map. This suggests that model performance is generally more sensitive to the quality and
characteristics of the input data than to the specific model architecture.

    Nonetheless, there are distinct cases where one of the models outperforms the other significantly. Wells where the single-
well model performs markedly better are shown in dark red, those where the global model performs better appear in dark blue.
These differences are likely related to the specific groundwater level dynamics at each site. This hypothesis is supported by
the observation that wells with similar $\Delta$NSE values, both negative and positive, often cluster spatially. This pattern suggests
that regional characteristics, potentially linked to hydrogeological conditions, influence whether the single-well or the global
model performs better in a given area.





# 5 Conclusions

Forecasting groundwater levels (GWL) remains a challenging task due to the complex and interconnected processes governing groundwater systems. Machine learning (ML) has shown great potential in addressing these challenges by capturing non-linear relationships in hydro-climatic data, even when observational data are sparse.

Despite significant progress in recent years, the field still lacks standardized, large-scale datasets. Most existing studies rely on localized and often inaccessible data, which limits reproducibility and hampers the transferability of results. To address

this gap, the present study introduces a comprehensive and standardized dataset for ML-based GWL modeling in Germany. It comprises 32 years of weekly groundwater level observations from over 3,000 monitoring wells, along with meteorological and site-specific static attributes. The dataset is publicly available and is intended to support systematic model comparisons, foster transparency and reproducibility, and promote further research through standardized documentation.

In addition, we provide three initial benchmark models: (i) single-well models, (ii) a global model using only dynamic

inputs, and (iii) a global model that incorporates both dynamic and static input features. These models serve as a starting point for future model development and evaluation.

We deliberately refrain from further analysis of the relationships between model performance and groundwater dynamics, hydrogeological conditions, or land use, as this lies beyond the scope of the current study. We leave it to future research to enhance the models using the provided benchmark dataset, with the hope that it will lead to valuable insights in the field of

ML-based groundwater level prediction.

# 6 Code and data availability

The complete GEMS-GER dataset is publicly available under an open-access license via Zenodo:
https://doi.org/10.5281/zenodo.15530171 (Ohmer et al., 2025). It includes groundwater level time series, meteorological and hydrological forcings, static site descriptors, and model performance metrics as described in this paper.

All associated code, documentation, and update announcements are maintained in the project's GitHub repository:
https://github.com/KITHydrogeology/GEMS-GER, ensuring transparency, traceability, and reproducibility.





**Table 1.** Overview of the dynamic climate variables used and their properties.

| Variable | Description | Unit | Resolution | Category | Wkly. Agg. | Source | Reference |
|---|---|---|---|---|---|---|---|
| HYRAS_tasmax | Max. temperature at 2m | °C | 1×1 km | Climate | mean | DWD HYRAS | (DWD, 2024a) |
| HYRAS_tas | Mean temperature at 2m | °C | 1×1 km | Climate | mean | DWD HYRAS | (DWD, 2024b) |
| HYRAS_tasmin | Min. temperature at 2m | °C | 1×1 km | Climate | mean | DWD HYRAS | (DWD, 2024c) |
| HYRAS_pr | Precipitation sum | mm | 1×1 km | Climate | sum | DWD HYRAS | (DWD, 2024d) |
| HYRAS_hurs | Relative humidity | % | 1×1 km | Climate | mean | DWD HYRAS | (DWD, 2024e) |
| DWD_evapo_p | Potential evapotranspiration | mm | 1×1 km | Climate | sum | DWD | (DWD-CDC, 2024a) |
| DWD_evapo_r | Actual evapotranspiration | mm | 1×1 km | Climate | sum | DWD | (DWD-CDC, 2024b) |
| DWD_evapo_fao | Reference evapotranspiration (FAO) | mm | 1×1 km | Climate | sum | DWD | (DWD-CDC, 2024c) |
| DWD_soil_moist | Soil moisture | % PAW | 1×1 km | Climate | mean | DWD | (DWD-CDC, 2024d) |
| DWD_soil_temp5cm | Soil temperature at 5cm depth | °C | 1×1 km | Climate | mean | DWD | (DWD-CDC, 2024e) |
| ERA5_sd | Mean snow depth | mm | 0.1×0.1 ° | Climate | mean | ERA5-Land | (Muñoz-Sabater et al., 2021) |
| ERA5_sm | Total snowmelt (m w.e.) | m | 0.1×0.1 ° | Climate | sum | ERA5-Land | (Muñoz-Sabater et al., 2021) |
| ERA5_sf | Total snowfall (m w.e.) | m | 0.1×0.1 ° | Climate | sum | ERA5-Land | (Muñoz-Sabater et al., 2021) |
| ERA5_sdwe | Snow depth (m w.e.) | m | 0.1×0.1 ° | Climate | mean | ERA5-Land | (Muñoz-Sabater et al., 2021) |
| ERA5_ssro | Sub-surface runoff sum | m | 0.1×0.1 ° | Hydrology | sum | ERA5-Land | (Muñoz-Sabater et al., 2021) |
| ERA5_sro | Runoff sum | m | 0.1×0.1 ° | Hydrology | sum | ERA5-Land | (Muñoz-Sabater et al., 2021) |



Table 2: Site-specific static environmental features

| Type | Variable | Description | Unit | Data Source/Reference |
|---|---|---|---|---|
| Hydrogeologic | GWN1000_GR | Mean Annual Groundwater Recharge of Germany at scale 1:1,000,000 (GWN1000) | mm | (BGR, 2019) |
| | HUEK250_HU | Hydrogeological Map of Germany at scale 1:250,000 (HÜK250), Hydrogeological Unit | cat. | (BGR and SGD, 2019) |
| | HUEK250_K | Hydrogeological Map of Germany at scale 1:250,000 (HÜK250), Hydraulic Conductivity | m/s | (BGR and SGD, 2019) |
| | HUEK250_RT | Hydrogeological Map of Germany at scale 1:250,000 (HÜK250), Rock Type | cat. | (BGR and SGD, 2019) |
| | HUEK250_CT | Hydrogeological Map of Germany at scale 1:250,000 (HÜK250), Type of Porosity | cat. | (BGR and SGD, 2019) |
| | HUEK250_DC | Hydrogeological Map of Germany at scale 1:250,000 (HÜK250), Degree of Consolidation | cat. | (BGR and SGD, 2019) |
| | HUEK250_GC | Hydrogeological Map of Germany at scale 1:250,000 (HÜK250), Geochemical Rock Type | cat. | (BGR and SGD, 2019) |
| | HYRAUM_HD | Hydrogeological Spatial Structure of Germany (HYRAUM), Hydrogeological District | cat. | (BGR and SGD, 2015) |
| | HYRAUM_MHD | Hydrogeological Spatial Structure of Germany (HYRAUM), Major Hydrogeological District | cat. | (BGR and SGD, 2015) |
| | HYSOG_SG | Global Hydrologic Soil Groups (HYSOGs250m) for Curve Number-Based Runoff Modeling | cat. | (ROSS et al., 2018) |
| | SWR_PR | Mean Annual Rate of Percolation from Soil in Germany (SWR1000) | mm | (BGR, 2003) |
| Hydrologic | EUMOHP_DSD_1-3 | Multiorder Hydrologic Position for Europe, Divide-to-Stream Distance (DSD) 1-3 | m | (Nölscher et al., 2022) |
| | EUMOHP_SD_1-3 | Multiorder Hydrologic Position for Europe, Lateral Position (LP) 1-3 | m | (Nölscher et al., 2022) |
| | EUMOHP_LP_1-3 | Multiorder Hydrologic Position for Europe, Stream Distance (SD) 1-3 | m | (Nölscher et al., 2022) |
| Soil | BUEK1000_RSA | Soil Map of Germany at scale 1:1,000,000 (BÜK1000), Reference Soil Association (RSA) | cat. | (BGR, 2020) |
| | HUMUS1000_OC | Organic Matter Content of Topsoils in Germany (HUMUS1000OB) | cat. | (BGR, 2007) |
| Landuse | CLC_90 | CORINE Land Cover 1990 (100 m), Europe, 6-yearly | cat. | (EEA, 2019a) |
| | CLC_00 | CORINE Land Cover 2000 (100 m), Europe, 6-yearly | cat. | (EEA, 2019b) |
| | CLC_06 | CORINE Land Cover 2006 (100 m), Europe, 6-yearly | cat. | (EEA, 2019c) |
| | CLC_12 | CORINE Land Cover 2012 (100 m), Europe, 6-yearly | cat. | (EEA, 2019d) |
| | CLC_18 | CORINE Land Cover 2018 (100 m), Europe, 6-yearly | cat. | (EEA, 2019e) |
| | MUNDIALIS_LU | Germany 2020 – Land cover classification based on Sentinel-2 data | cat. | (Riembauer et al., 2021) |
| Terrain Geomorphology | GMK1000_GU | Geomorphographic Map of Germany 1:1,000,000, Geomorphic Unit | cat. | (BGR, 2014) |
| | DTM20_FD | Topographical Flow Directions (SGD, 2024), computed on DEM 20 m | - | (SGD, 2024) |
| | DTM20_SL | Slope, computed with Module Slope, Aspect, Curvature on DEM 20 m | ° | (SGD, 2024) |
| | DTM20_AS | Aspect, computed with Module Slope, Aspect, Curvature on DEM 20 m | ° | (SGD, 2024) |
| | DTM20_GC | General Curvature, computed with Module Slope, Aspect, Curvature on DEM 20 m | - | (SGD, 2024) |
| | DTM20_PLC | Plan Curvature, computed with Module Slope, Aspect, Curvature on DEM 20 m | - | (SGD, 2024) |
| | DTM20_PRC | Profile Curvature, computed with Module Slope, Aspect, Curvature on DEM 20 m | - | (SGD, 2024) |



| Type | Variable | Description | Unit | Data Source/Reference |
|---|---|---|---|---|
| | DTM20_FA | Flow Accumulation, computed with Tool Flow Accumulation (Parallelizable) on DEM 20 m | n cells | (SGD, 2024) |
| | DTM20_TRI | Terrain Ruggedness Index (TRI) computed with Module Terrain Ruggedness Index on DEM 20 m | - | (SGD, 2024) |
| | DTM20_CI | Convergence/divergence index computed with Tool Convergence Index on DEM 20 m | - | (SGD, 2024) |
| | DTM20_MRI | Melton Ruggedness Number computed with Module Melton Ruggedness Number on DEM 20 m | - | (SGD, 2024) |
| MW-Metadata | MW_ID | Monitoring Well ID | ID | |
| | Proj_ID | Original Monitoring ID (from authorities) | ID | |
| | Operator | Network Operator (environmental agencies of German federal states) | cat. | |
| | Easting (3035) | Easting (EPSG:3035) | m | |
| | Northing (3035) | Northing (EPSG:3035) | m | |
| | Elevation | Ground Surface Elevation (m above sea level, m asl) | m | |
| | Depth | Total Well Depth (m below ground level, m bgl) | m | |
| | UpFilter | Top of Screen (m below ground level, m bgl) | m | |
| | LoFilter | Bottom of Screen (m below ground level, m bgl) | m | |
| | ScrLength | Screen Length (LoFilter - UpFilter) | m | |
| | AquiferMed | Aquifer Medium ('Fractured', 'Porous', 'Unknown', 'Karstic') | cat. | |
| | PreState | Pressure Condition ('Confined', 'Unconfined', 'Unknown') | cat. | |





**Table 3.** Overview of the model results.

| Model | NSE min | NSE mean | NSE median | NSE max |
|---|---|---|---|---|
| Single | -3.47 | 0.41 | 0.52 | 0.94 |
| Global Dyn only | -6.78 | 0.32 | 0.44 | 0.88 |
| Global Dyn + Stat | -4.31 | 0.39 | 0.50 | 0.91 |
| | RMSE min | RMSE mean | RMSE median | RMSE max |
| Single | 0.03 | 0.40 | 0.28 | 7.67 |
| Global Dyn only | 0.03 | 0.43 | 0.30 | 6.93 |
| Global Dyn + Stat | 0.02 | 0.42 | 0.29 | 7.17 |
| | $R^2$ min | $R^2$ mean | $R^2$ median | $R^2$ max |
| Single | < 0.01 | 0.48 | 0.53 | 0.93 |
| Global Dyn only | <0.01 | 0.46 | 0.47 | 0.91 |
| Global Dyn + Stat | < 0.01 | 0.49 | 0.53 | 0.93 |
| | Bias min | Bias mean | Bias median | Bias max |
| Single | -4.44 | 0.03 | 0.01 | 6.34 |
| Global Dyn only | -5.29 | -0.01 | -0.01 | 5.51 |
| Global Dyn + Stat | -5.21 | 0.01 | 0.01 | 5.80 |
| | No. NSE $\leq$ 0 | No. $0 <$ NSE $\leq 0.5$ | No. $0.5 <$ NSE $\leq 0.8$ | No. NSE $> 0.8$ |
| Single | 404 | 1134 | 1478 | 191 |
| Global Dyn only | 471 | 1466 | 1241 | 29 |
| Global Dyn + Stat | 396 | 1228 | 1471 | 112 |



**Figure 1.** Data preprocessing steps for the GEMS-GER groundwater dataset, including harmonization, completeness filtering, detection of abrupt shifts (PELT), multi-method outlier identification, and iterative imputation. Only plausibility-checked and high-quality series were retained (n = 3,207). Each panel in the figure shows the state of the dataset at a specific processing step. Each row in the heatmaps represents one standardized groundwater time series from a monitoring well, with the y-axis corresponding to time (1991–2023). Blue indicates high groundwater levels, red indicates low levels, and black marks missing values.

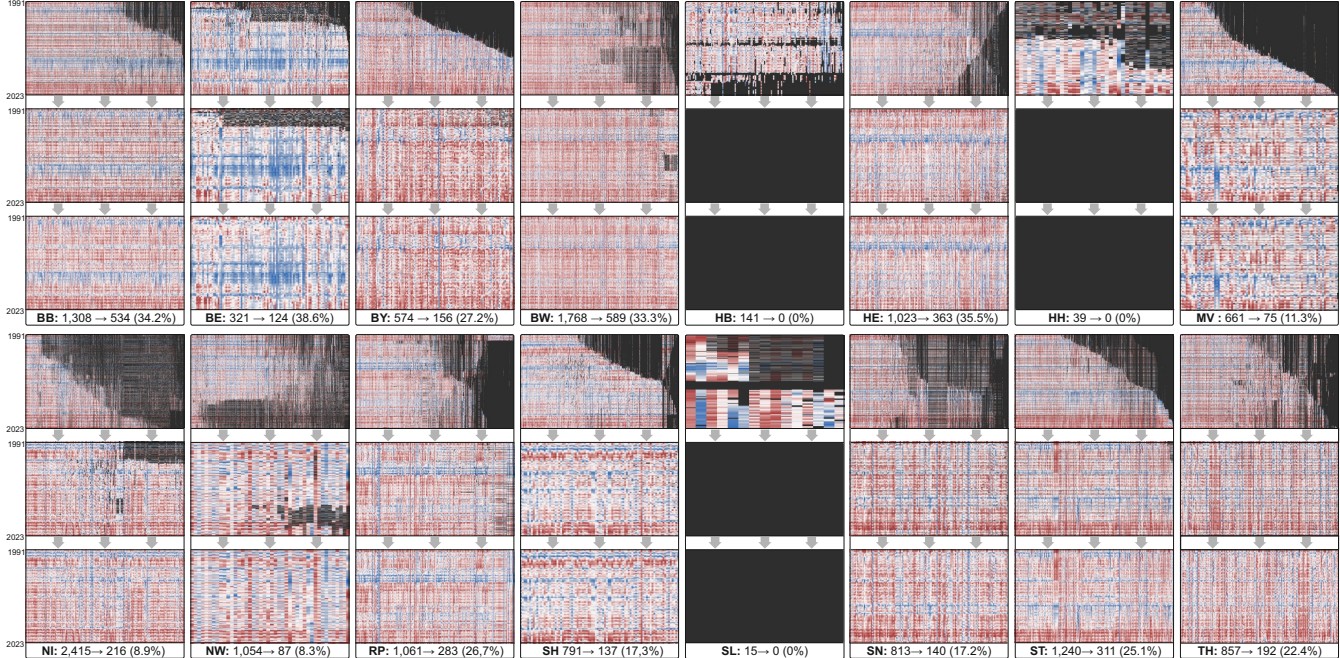

**Figure 2.** Data availability by federal state after (top) Section 2.2.1 Data Integration and Weekly Aggregation, (middle) Section 2.2.2 Data Gap Filtering, and (bottom) Section 2.2.5 Data Imputation. Each row in the heatmaps represents one standardized groundwater level time series from a monitoring well, with the y-axis corresponding to time (1991–2023). Blue indicates high groundwater levels, red indicates low levels, and black marks missing values. **State abbreviations:** BB – Brandenburg, BE – Berlin, BW – Baden-Württemberg, BY – Bavaria, HB – Bremen, HE – Hesse, HH – Hamburg, MV – Mecklenburg-Western Pomerania, NI – Lower Saxony, NW – North Rhine-Westphalia, RP – Rhineland-Palatinate, SL – Saarland, SN – Saxony, ST – Saxony-Anhalt, SH – Schleswig-Holstein, TH – Thuringia.

**Figure 3.** Spatial distribution of the 3,207 groundwater monitoring wells across the ten Major Hydrological Districts (MHDs) in Germany: (1) North and Central German Unconsolidated Rock District, (2) Rhenish-Westphalian Lowland, (3) Upper Rhine Graben with Mainz Basin and North Hessian Tertiary, (4) Alpine Foreland, (5) Central German Fault-block Land, (6) West and South German Scarplands and Fault-block Land, (7) Alps, (8) West and Central German Basement, (9) Southeast German Basement, (10) Southwest German Basement. The violin plots show the distribution of seven dynamic indicators across these regions, as described by Wunsch et al. (2022b) and Richter et al. (1996): *SD_diff* (short-term variability), *range_ratio* (interannual vs. total variability), *ex_vals* (frequency of peaks), *seasonal_behaviour* (fit to annual cycle), *periodicity* (weekly pattern recurrence), *yearly_variance* (amplitude of seasonal fluctuations), and *HPD* (persistence of high groundwater levels).



**Figure 4.** Extracts of the dataset used in the study, showing dynamic variables (mean groundwater level 1991–2022, mean annual precipitation, and potential evapotranspiration) and selected static raster layers (soil group, porosity type, hydrological region, organic matter content, land use, and groundwater recharge).



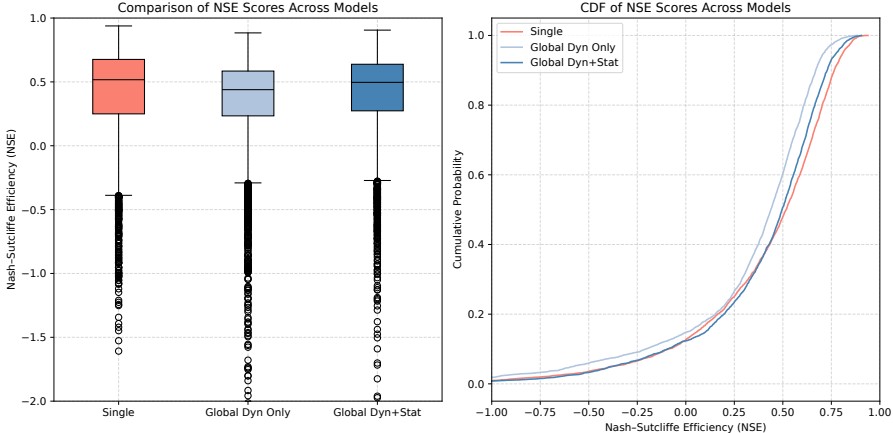

**Figure 5.** Comparison of NSE scores across model variants, as boxplots and CDF. For plotting reasons, the lower limits of values are set to -2 and -1, respectively, so that some outliers are not shown.



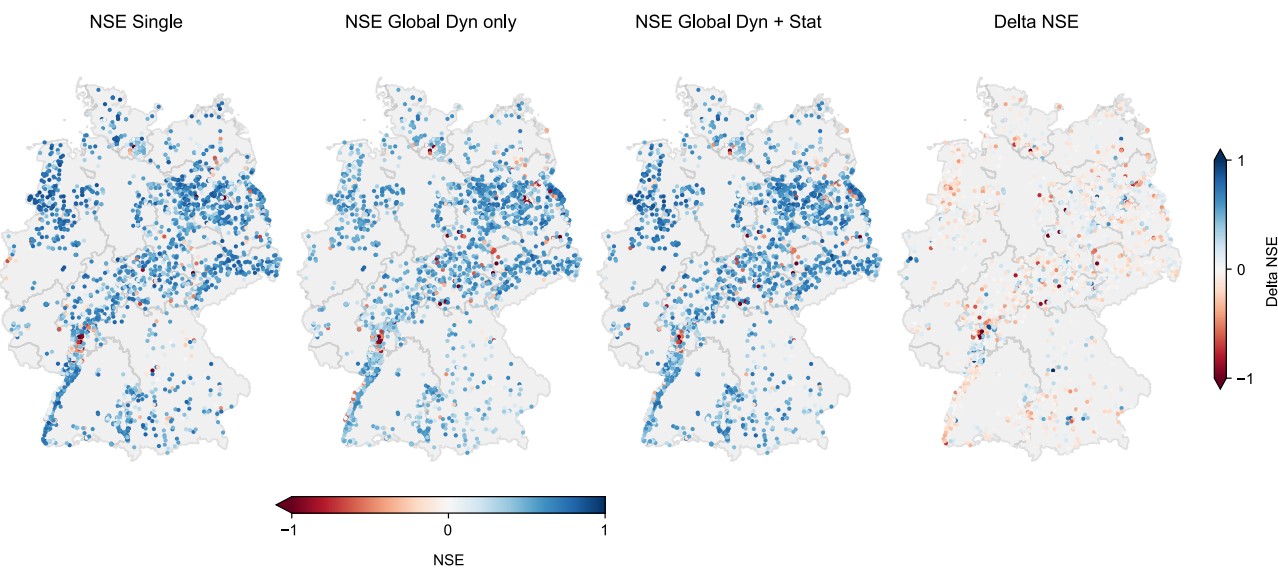

**Figure 6.** NSE values for all wells and model variants, and the delta NSE between the single well and global model with dynamic and static values (positive/blue values indicate that the global model is better, negative/red values indicate that the single well model is better).

## Appendix: Appendix A

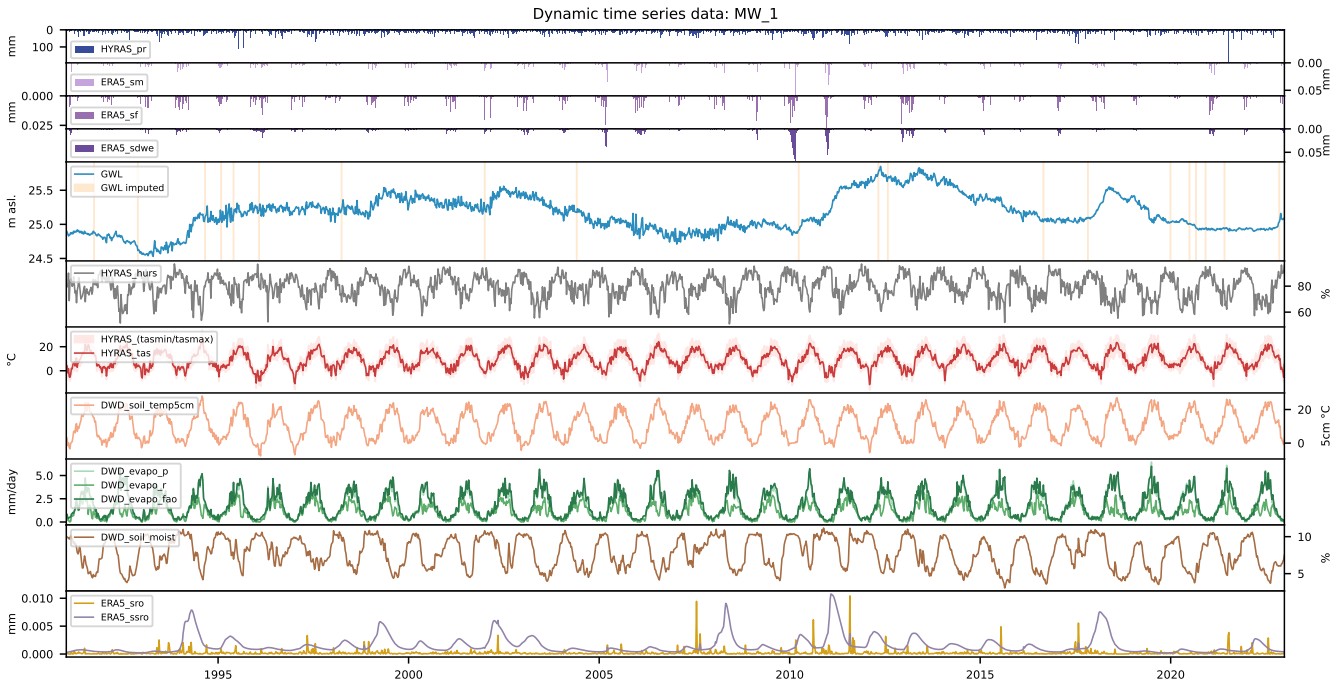

**Figure A1.** An example plot of selected dynamic time series (groundwater level, precipitation, temperature, evapotranspiration, and runoff) is shown for Monitoring Well MW_1. Corresponding illustrations for all wells are included in the dataset in the `GEMS-GER_figures/` directory.

*Author contributions.* MO wrote most of the manuscript, with contributions from TL, especially on the modelling part. MO did the data preparation and processing, with contributions from BHE, BHA, MGO and MN. TL conceptualised and calculated the benchmark models. SB took care of the procurement of the groundwater level data from the federal institutions, assisted by PC. All authors contributed to discussions on the methodology and performed review and editing tasks.

*Competing interests.* The authors declare that they have no conflict of interest.

*Acknowledgements.* We would like to thank all employees of the responsible authorities, who were involved in the data collection or provision of the data. Without their help, the creation of such an extensive and valuable dataset would not have been possible.




All programming was done in Python version 3.9 (van Rossum, 1995) and the associated libraries, including NumPy (Harris et al., 2020), Pandas (McKinney, 2010), Tensorflow (Abadi et al., 2016), Keras (Chollet, 2015), SciPy (Virtanen et al., 2020), Scikit-learn (Pedregosa et al., 2011b) and Matplotlib (Hunter, 2007).

The authors further acknowledge support by the state of Baden-Württemberg through bwHPC.



## Appendix: References

Abadi, M., Agarwal, A., Barham, P., Brevdo, E., Chen, Z., Citro, C., Corrado, G. S., Davis, A., Dean, J., Devin, M., Ghemawat, S., Good-
fellow, I., Harp, A., Irving, G., Isard, M., Jia, Y., Jozefowicz, R., Kaiser, , Kudlur, M., Levenberg, J., Mané, D., Monga, R., Moore, S.,
Murray, D., Olah, C., Schuster, M., Shlens, J., Steiner, B., Sutskever, I., Talwar, K., Tucker, P., Vanhoucke, V., Vasudevan, V., Viégas, F.,
Vinyals, O., Warden, P., Wattenberg, M., Wicke, M., Yu, Y., and Zheng, X.: TensorFlow: Large-Scale Machine Learning on Heterogeneous
Distributed Systems, arXiv preprint arXiv:1603.04467, 2016.

Addor, N., Newman, A. J., Mizukami, N., and Clark, M. P.: The CAMELS data set: catchment attributes and meteorology for large-sample
studies, Hydrology and Earth System Sciences, 21, 5293–5313, https://doi.org/10.5194/hess-21-5293-2017, 2017.

Ahmadi, A., Olyaei, M., Heydari, Z., Emami, M., Zeynolabedin, A., Ghomlaghi, A., Daccache, A., Fogg, G. E., and Sadegh, M.: Groundwater
Level Modeling with Machine Learning: A Systematic Review and Meta-Analysis, Water, 14, 949, https://doi.org/10.3390/w14060949,
2022.

Barzegar, R., Fijani, E., Asghari Moghaddam, A., and Tziritis, E.: Forecasting of groundwater level fluctuations us-
ing ensemble hybrid multi-wavelet neural network-based models, Science of The Total Environment, 599-600, 20–31,
https://doi.org/10.1016/j.scitotenv.2017.04.189, 2017.

BGR: Mean Annual Rate of Percolation from the Soil in Germany (SWR1000_250), https://services.bgr.de/arcgis/rest/services/boden/
swr1000/MapServer/0, 2003.

BGR: Organic Matter Content of Top-Soils in Germany HUMUS1000OB, 2007.

BGR: Geomorphographic Map of Germany 1:1,000,000 (GMK1000R), https://gdk.gdi-de.org/geonetwork/srv/api/records/
60ab5e4e-9493-44b0-9cae-d9ce603de742, 2014.

BGR: Mean Annual Groundwater Recharge of Germany 1:1,000,000 (GWN1000), https://geoportal.bgr.de/mapapps/resources/apps/
geoportal/index.html?lang=de#/geoviewer?metadataId=40E14FF1-99D4-43DA-AF7B-C039F0463BF8, 2019.

BGR: Digital soil map of Germany 1 : 1,000,000 (BUEK 1000), https://services.bgr.de/wms/boden/buek1000de/?, 2020.

BGR and SGD: Hydrogeological spatial structure of Germany (HYRAUM), https://www.bgr.bund.de/DE/Themen/Wasser/Projekte/
abgeschlossen/Beratung/Hyraum/hyraum_grossraeume+raeume_karte.html, 2015.

BGR and SGD: Hydrogeological Map of Germany 1:250,000 (HÜK250), https://www.bgr.bund.de/DE/Themen/Wasser/Projekte/laufend/
Beratung/Huek200/huek200_projektbeschr.html, 2019.

Breunig, M. M., Kriegel, H.-P., Ng, R. T., and Sander, J.: LOF: identifying density-based local outliers, ACM SIGMOD Record, 29, 93–104,
https://doi.org/10.1145/335191.335388, 2000.

Buck, S. F.: A Method of Estimation of Missing Values in Multivariate Data Suitable for Use with an Electronic Computer, Journal of the
Royal Statistical Society Series B: Statistical Methodology, 22, 302–306, https://doi.org/10.1111/j.2517-6161.1960.tb00375.x, 1960.

Buuren, S. V. and Groothuis-Oudshoorn, K.: **mice** : Multivariate Imputation by Chained Equations in *R*, Journal of Statistical Software, 45,
https://doi.org/10.18637/jss.v045.i03, 2011.

Chen, C., He, W., Zhou, H., Xue, Y., and Zhu, M.: A comparative study among machine learning and numerical models for simulating
groundwater dynamics in the Heihe River Basin, northwestern China, Scientific Reports, 10, 3904, https://doi.org/10.1038/s41598-020-
60698-9, 2020.

Chen, L.-H., Chen, C.-T., and Pan, Y.-G.: Groundwater Level Prediction Using SOM-RBFN Multisite Model, Journal of Hydrologic Engi-
neering, 15, 624–631, https://doi.org/10.1061/(ASCE)HE.1943-5584.0000218, 2010.



Chollet, F.: Keras, https://github.com/keras-team/keras, accessed: 2025-05-28, 2015.

Coulibaly, P., Anctil, F., Aravena, R., and Bobée, B.: Artificial neural network modeling of water table depth fluctuations, Water Resources Research, 37, 885–896, https://doi.org/10.1029/2000WR900368, 2001.

De Graaf, I. E. M., Gleeson, T., (Rens) Van Beek, L. P. H., Sutanudjaja, E. H., and Bierkens, M. F. P.: Environmental flow limits to global groundwater pumping, Nature, 574, 90–94, https://doi.org/10.1038/s41586-019-1594-4, 2019.

Destatis: Wassergewinnung: Die regionale Zuordnung erfolgt über den Standort des Wasserversorgungsunternehmens, https://www-genesis.destatis.de/datenbank/online/table/32211-0002, 2025.

DWD: Raster data set of daily maximum temperature in °C for Germany – HYRAS-DE-TASMAX, Version v6.0, https://opendata.dwd.de/climate_environment/CDC/grids_germany/daily/hyras_de/air_temperature_max/, 2024a.

DWD: Raster data set of mean temperature in °C for Germany – HYRAS-DE-TAS, Version v6.0, https://opendata.dwd.de/climate_environment/CDC/grids_germany/daily/hyras_de/air_temperature_mean/, 2024b.

DWD: Raster data set of minimum temperature in °C for Germany – HYRAS-DE-TASMIN, Version v6.0, https://opendata.dwd.de/climate_environment/CDC/grids_germany/daily/hyras_de/air_temperature_min/, 2024c.

DWD: Raster data set of precipitation sums in mm for Germany – HYRAS-DE-PR, Version v6.0, https://opendata.dwd.de/climate_environment/CDC/grids_germany/daily/hyras_de/precipitation_sum/, 2024d.

DWD: Raster data set of mean relative humidity in % for Germany – HYRAS-DE-HURS, Version v6.0, https://opendata.dwd.de/climate_environment/CDC/grids_germany/daily/hyras_de/humidity/, 2024e.

DWD-CDC: Daily grids of potential evapotranspiration over grass, version 0.x, https://opendata.dwd.de/climate_environment/CDC/grids_germany/daily/evapo_p/, 2024a.

DWD-CDC: Daily grids of actual evapotranspiration over grass and sandy loam, version 0.x, https://opendata.dwd.de/climate_environment/CDC/grids_germany/daily/evapo_r/, 2024b.

DWD-CDC: Daily grids of FAO Grass reference evapotranspiration, Version v1.0, https://opendata.dwd.de/climate_environment/CDC/grids_germany/daily/evapo_fao/, 2024c.

DWD-CDC: Daily grids of soil moisture under grass and sandy loam, version 0.x, https://opendata.dwd.de/climate_environment/CDC/grids_germany/daily/soil_moist/, 2024d.

DWD-CDC: Daily grids of soil temperature at 5cm depth, version 0.x, https://opendata.dwd.de/climate_environment/CDC/grids_germany/daily/soil_temp/, 2024e.

EEA: CORINE Land Cover 1990 (raster 100 m), Europe, 6-yearly - version 2020_20u1, May 2020, https://doi.org/10.2909/C89324EF-7729-4477-9F1B-623F5F88EAA1, 2019a.

EEA: CORINE Land Cover 2000 (raster 100 m), Europe, 6-yearly - version 2020_20u1, May 2020, https://doi.org/10.2909/DDACBD5E-068F-4E52-A596-D606E8DE7F40, 2019b.

EEA: CORINE Land Cover 2006 (raster 100 m), Europe, 6-yearly - version 2020_20u1, May 2020, https://doi.org/10.2909/08560441-2FD5-4EB9-BF4C-9EF16725726A, 2019c.

EEA: CORINE Land Cover 2012 (raster 100 m), Europe, 6-yearly - version 2020_20u1, May 2020, https://doi.org/10.2909/A84AE124-C5C5-4577-8E10-511BFE55CC0D, 2019d.

EEA: CORINE Land Cover 2018 (raster 100 m), Europe, 6-yearly - version 2020_20u1, May 2020, https://doi.org/10.2909/960998C1-1870-4E82-8051-6485205EBBAC, 2019e.



Emamgholizadeh, S., Moslemi, K., and Karami, G.: Prediction the Groundwater Level of Bastam Plain (Iran) by Artificial Neural Network (ANN) and Adaptive Neuro-Fuzzy Inference System (ANFIS), Water Resources Management, 28, 5433–5446, https://doi.org/10.1007/s11269-014-0810-0, 2014.

Feng, F., Ghorbani, H., and Radwan, A. E.: Predicting groundwater level using traditional and deep machine learning algorithms, Frontiers in Environmental Science, 12, 1291 327, https://doi.org/10.3389/fenvs.2024.1291327, 2024.

Gomez, M., Nölscher, M., Hartmann, A., and Broda, S.: Assessing groundwater level modelling using a 1-D convolutional neural network (CNN): linking model performances to geospatial and time series features, Hydrology and Earth System Sciences, 28, 4407–4425, https://doi.org/10.5194/hess-28-4407-2024, 2024.

Guzman, S. M., Paz, J. O., Tagert, M. L. M., and Mercer, A. E.: Evaluation of Seasonally Classified Inputs for the Prediction of Daily Groundwater Levels: NARX Networks Vs Support Vector Machines, Environmental Modeling & Assessment, 24, 223–234, https://doi.org/10.1007/s10666-018-9639-x, 2019.

Han, Z., Li, F., Zhao, Y., and Liu, C.: Investigation into groundwater level prediction within a deep learning framework: Incorporating the spatial dynamics of adjacent wells, Journal of Hydrology, 657, 133 097, https://doi.org/10.1016/j.jhydrol.2025.133097, 2025.

Harris, C. R., Millman, K. J., van der Walt, S. J., Gommers, R., Virtanen, P., Cournapeau, D., Wieser, E., Taylor, J., Berg, S., Smith, N. J., Kern, R., Picus, M., Hoyer, S., van Kerkwijk, M. H., Brett, M., Haldane, A., Del Río, J. F., Wiebe, M., Peterson, P., Gérard-Marchant, P., Sheppard, T., Reddy, D., Weckesser, W., Abbasi, H., Gohlke, C., and Oliphant, T. E.: Array programming with NumPy, Nature, 585, 357–362, https://doi.org/10.1038/s41586-020-2649-2, 2020.

Heudorfer, B., Liesch, T., and Broda, S.: On the challenges of global entity-aware deep learning models for groundwater level prediction, Hydrology and Earth System Sciences, 28, 525–543, https://doi.org/10.5194/hess-28-525-2024, 2024.

Huang, F., Huang, J., Jiang, S.-H., and Zhou, C.: Prediction of groundwater levels using evidence of chaos and support vector machine, Journal of Hydroinformatics, 19, 586–606, https://doi.org/10.2166/hydro.2017.102, 2017.

Hunter, J. D.: Matplotlib: A 2D Graphics Environment, Computing in Science & Engineering, 9, 90–95, https://doi.org/10.1109/MCSE.2007.55, 2007.

Kasiviswanathan, K. S., Saravanan, S., Balamurugan, M., and Saravanan, K.: Genetic programming based monthly groundwater level forecast models with uncertainty quantification, Modeling Earth Systems and Environment, 2, 27, https://doi.org/10.1007/s40808-016-0083-0, 2016.

Kholghi, M. and Hosseini, S. M.: Comparison of Groundwater Level Estimation Using Neuro-fuzzy and Ordinary Kriging, Environmental Modeling & Assessment, 14, 729–737, https://doi.org/10.1007/s10666-008-9174-2, 2009.

Killick, R., Fearnhead, P., and Eckley, I. A.: Optimal detection of changepoints with a linear computational cost, Journal of the American Statistical Association, 107, 1590–1598, https://doi.org/10.1080/01621459.2012.737745, arXiv:1101.1438 [stat], 2012.

Kratzert, F., Nearing, G., Addor, N., Frame, J. M., Gauch, M., Xu, Y., Mai, J., Zeng, Z., Hochreiter, S., Gupta, H., Klotz, D., Klambauer, G., Pfister, L., Reichstein, M., Shen, C., Wood, A., Rakovec, O., Newman, A. J., Clark, M. P., Lerat, J., Andréassian, V., Mendoza, P., Coxon, G., Mizukami, N., Smith, T., Westra, S., Gharari, S., Nearing, G., Duan, Q., Burek, P., and Hall, J.: Caravan – A global community dataset for large-sample hydrology, Scientific Data, 10, 61, https://doi.org/10.1038/s41597-023-01975-w, 2023.

Krishna, B., Satyaji Rao, Y. R., and Vijaya, T.: Modelling groundwater levels in an urban coastal aquifer using artificial neural networks, Hydrological Processes, 22, 1180–1188, https://doi.org/10.1002/hyp.6686, 2008.

Lallahem, S., Mania, J., Hani, A., and Najjar, Y.: On the use of neural networks to evaluate groundwater levels in fractured media, Journal of Hydrology, 307, 92–111, https://doi.org/10.1016/j.jhydrol.2004.10.005, 2005.





Liu, F. T., Ting, K. M., and Zhou, Z.-H.: Isolation Forest, in: 2008 Eighth IEEE International Conference on Data Mining, pp. 413–422, IEEE, Pisa, Italy, ISBN 978-0-7695-3502-9, https://doi.org/10.1109/ICDM.2008.17, 2008.

Liu, F. T., Ting, K. M., and Zhou, Z.-H.: Isolation-Based Anomaly Detection, ACM Transactions on Knowledge Discovery from Data, 6, 1–39, https://doi.org/10.1145/2133360.2133363, 2012.

MacKay, D. J. C.: Bayesian Interpolation, Neural Computation, 4, 415–447, https://doi.org/10.1162/neco.1992.4.3.415, 1992.

McKinney, W.: Data Structures for Statistical Computing in Python, in: Proceedings of the 9th Python in Science Conference, edited by van der Walt, S. and Millman, J., pp. 51–56, 2010.

Moosavi, V., Vafakhah, M., Shirmohammadi, B., and Behnia, N.: A Wavelet-ANFIS Hybrid Model for Groundwater Level Forecasting for Different Prediction Periods, Water Resources Management, 27, 1301–1321, https://doi.org/10.1007/s11269-012-0239-2, 2013.

Muñoz-Sabater, J., Dutra, E., Agustí-Panareda, A., Albergel, C., Arduini, G., Balsamo, G., Boussetta, S., Choulga, M., Harrigan, S., Hersbach, H., Martens, B., Miralles, D. G., Piles, M., Rodríguez-Fernández, N. J., Zsoter, E., Buontempo, C., and Thépaut, J.-N.: ERA5-Land: a state-of-the-art global reanalysis dataset for land applications, Earth System Science Data, 13, 4349–4383, https://doi.org/10.5194/essd-13-4349-2021, 2021.

Nayak, P. C., Rao, Y. R. S., and Sudheer, K. P.: Groundwater Level Forecasting in a Shallow Aquifer Using Artificial Neural Network
Approach, Water Resources Management, 20, 77–90, https://doi.org/10.1007/s11269-006-4007-z, 2006.

Nölscher, M., Mutz, M., and Broda, S.: Multiorder hydrologic Position for Europe — a Set of Features for Machine Learning and Analysis in Hydrology, Scientific Data, 9, 662, https://doi.org/10.1038/s41597-022-01787-4, 2022.

Ohmer, M., Liesch, T., Habbel, B., Heudorfer, B., Gomez, M., Clos, P., Nölscher, M., and Broda, S.: GEMS-GER: A Machine Learning Benchmark Dataset of Long-Term Groundwater Levels, Environment, Meteorology, Site Properties,
https://doi.org/10.5281/zenodo.15530171, zenodo repository, Version v3 (August 4, 2025), 2025.

Pedregosa, F., Varoquaux, G., Gramfort, A., Michel, V., Thirion, B., Grisel, O., Blondel, M., Prettenhofer, P., Weiss, R., Dubourg, V., Passos, A., Vanderplas, J., Cournapeau, D., Perrot, M., Perrot, E., and Brucher, M.: Scikit-learn: Machine Learning in Python, Journal of Machine Learning Research, 12, 2825–2830, 2011a.

Pedregosa, F., Varoquaux, G., Gramfort, A., Michel, V., Thirion, B., Grisel, O., Blondel, M., Prettenhofer, P., Weiss, R., Dubourg, V.,
Vanderplas, J., Passos, A., Cournapeau, D., Brucher, M., Perrot, M., and Duchesnay, : Scikit-learn: Machine Learning in Python, Journal of Machine Learning Research, 12, 2825–2830, 2011b.

Richter, B. D., Baumgartner, J. V., Powell, J., and Braun, D. P.: A Method for Assessing Hydrologic Alteration within Ecosystems, Conservation Biology, 10, 1163–1174, https://doi.org/10.1046/j.1523-1739.1996.10041163.x, 1996.

Riembauer, G., Weinmann, A., Xu, S., Eichfuss, S., Eberz, C., and Neteler, M.: Germany-wide Sentinel-2 based land cover classification and
510 change detection for settlement and infrastructure monitoring, Publications Office, LU, proceedings of the 2021 conference on big data from space, virtual. edn., https://data.europa.eu/doi/10.2760/125905, 2021.

ROSS, C., PRIHODKO, L., ANCHANG, J., KUMAR, S., JI, W., and HANAN, N.: Global Hydrologic Soil Groups (HYSOGs250m) for Curve Number-Based Runoff Modeling, https://doi.org/10.3334/ORNLDAAC/1566, artwork Size: 571.82448 MB Pages: 571.82448 MB, 2018.

Sadat-Noori, M., Glamore, W., and Khojasteh, D.: Groundwater level prediction using genetic programming: the importance of precipitation data and weather station location on model accuracy, Environmental Earth Sciences, 79, 37, https://doi.org/10.1007/s12665-019-8776-0, 2020.





Samani, S., Vadiati, M., Nejatijahromi, Z., Etebari, B., and Kisi, O.: Groundwater level response identification by hybrid wavelet–machine learning conjunction models using meteorological data, Environmental Science and Pollution Research, 30, 22 863–22 884, https://doi.org/10.1007/s11356-022-23686-2, 2022.

Saumen, M. and Tiwari, R. K.: A comparative study of artificial neural networks, Bayesian neural networks and adaptive neuro-fuzzy inference system in groundwater level prediction., 71, 3147–3160, https://doi.org/10.1007/s12665-013-2702-7, place: Heidelberg Publisher: Springer Berlin, 2014.

Seabold, S. and Perktold, J.: Statsmodels: Econometric and Statistical Modeling with Python, pp. 92–96, Austin, Texas, https://doi.org/10.25080/Majora-92bf1922-011, 2010.

SGD: OpenDTM-DE: 1-meter Digital Terrain Models of all 16 German federal states [Data set]. The datasets were aggregated and made publicly accessible via the OpenDEM, https://www.opendem.info/, the datasets are licensed under open data terms, specifically: Data License Germany – Attribution – Version 2.0, Creative Commons Attribution 4.0 International, Data License Germany – Zero – Version 2.0, and the custom open data terms of the state of Hesse (https://hvbg.hessen.de/open-data), 2024.

Shaikh, M. and Birajdar, F.: Artificial intelligence in groundwater management: Innovations, challenges, and future prospects, International Journal of Science and Research Archive, 11, 502–512, https://doi.org/10.30574/ijsra.2024.11.1.0105, 2024.

Solgi, R., Loáiciga, H. A., and Kram, M.: Long short-term memory neural network (LSTM-NN) for aquifer level time series forecasting using in-situ piezometric observations, Journal of Hydrology, 601, 126 800, https://doi.org/10.1016/j.jhydrol.2021.126800, 2021.

Tao, H., Hameed, M. M., Marhoon, H. A., Zounemat-Kermani, M., Heddam, S., Kim, S., Sulaiman, S. O., Tan, M. L., Sa'adi, Z., Mehr, A. D., Allawi, M. F., Abba, S., Zain, J. M., Falah, M. W., Jamei, M., Bokde, N. D., Bayatvarkeshi, M., Al-Mukhtar, M., Bhagat, S. K., Tiyasha, T., Khedher, K. M., Al-Ansari, N., Shahid, S., and Yaseen, Z. M.: Groundwater level prediction using machine learning models: A comprehensive review, Neurocomputing, 489, 271–308, https://doi.org/10.1016/j.neucom.2022.03.014, 2022.

Tipping, M. E.: Sparse bayesian learning and the relevance vector machine, 2001.

Truong, C., Oudre, L., and Vayatis, N.: Selective review of offline change point detection methods, Signal Processing, 167, 107 299, https://doi.org/10.1016/j.sigpro.2019.107299, 2020.

van Rossum, G.: Python Programming Language, Version 3.9, https://www.python.org, accessed: 2025-05-28, 1995.

Virtanen, P., Gommers, R., Oliphant, T. E., Haberland, M., Reddy, T., Cournapeau, D., Burovski, E., Peterson, P., Weckesser, W., Bright, J., van der Walt, S. J., Brett, M., Wilson, J., Millman, K. J., Mayorov, N., Nelson, A. R. J., Jones, E., Kern, R., Larson, E., Carey, C., Polat, , Feng, Y., Moore, E. W., VanderPlas, J., Laxalde, D., Perktold, J., Cimrman, R., Henriksen, I., Quintero, E. A., Harris, C. R., Archibald, A. M., Ribeiro, A. H., Pedregosa, F., and van Mulbregt, P.: SciPy 1.0: Fundamental Algorithms for Scientific Computing in Python, Nature Methods, 17, 261–272, https://doi.org/10.1038/s41592-019-0686-2, 2020.

Wunsch, A., Liesch, T., and Broda, S.: Groundwater level forecasting with artificial neural networks: a comparison of long short-term memory (LSTM), convolutional neural networks (CNNs), and non-linear autoregressive networks with exogenous input (NARX), Hydrology and Earth System Sciences, 25, 1671–1687, https://doi.org/10.5194/hess-25-1671-2021, 2021.

Wunsch, A., Liesch, T., and Broda, S.: Deep learning shows declining groundwater levels in Germany until 2100 due to climate change, Nature Communications, 13, 1221, https://doi.org/10.1038/s41467-022-28770-2, 2022a.

Wunsch, A., Liesch, T., and Broda, S.: Feature-based Groundwater Hydrograph Clustering Using Unsupervised Self-Organizing Map-Ensembles, Water Resources Management, 36, 39–54, https://doi.org/10.1007/s11269-021-03006-y, 2022b.

Yang, X. and Zhang, Z.: A CNN-LSTM Model Based on a Meta-Learning Algorithm to Predict Groundwater Level in the Middle and Lower Reaches of the Heihe River, China, Water, 14, 2377, https://doi.org/10.3390/w14152377, 2022.





Ying, Z., Wenxi, L., Haibo, C., and Jiannan, L.: Comparison of three forecasting models for groundwater levels: a case study in the semiarid area of west Jilin Province, China, Journal of Water Supply: Research and Technology-Aqua, 63, 671–683, https://doi.org/10.2166/aqua.2014.023, _eprint: https://iwaponline.com/aqua/article-pdf/63/8/671/400008/671.pdf, 2014.

Yoon, H., Jun, S.-C., Hyun, Y., Bae, G.-O., and Lee, K.-K.: A comparative study of artificial neural networks and support vector machines
for predicting groundwater levels in a coastal aquifer, Journal of Hydrology, 396, 128–138, https://doi.org/10.1016/j.jhydrol.2010.11.002, 2011.