# Peer review of "GEMS-GER: A Machine Learning Benchmark Dataset of Long-Term Groundwater Levels in Germany with Meteorological Forcings and Site-Specific Environmental Features"

_Earth System Science Data, 2025_

## Author Comment (AC1)

The manuscript "GEMS-GER: A Machine Learning Benchmark Dataset of Long-Term Groundwater Levels in Germany with Meteorological Forcings and Site-Specific Environmental Features" presents a valuable new groundwater dataset for Germany that consolidates scattered information into a unified and accessible format. The dataset includes time series of groundwater levels, meteorological forcings, and a range of static environmental descriptors. The authors have clearly invested considerable effort in collecting, harmonizing, and cleaning these data, and the result is a resource of great potential value for the hydrological and environmental science communities. Overall, I find this paper well-prepared and the dataset highly relevant. It is very positive to note that the documentation and data availability is clear and everything is easy to follow. This is a great example of how things should ideally always be. However, I believe some revisions are necessary to further strengthen the manuscript. My comments are as follows:

- We sincerely thank Reviewer 1 for the thorough and encouraging evaluation of our manuscript. We greatly appreciate the reviewer's recognition of the dataset's scientific value, the clarity of the documentation, and the overall quality of data preparation.
- The reviewer's constructive remarks have helped us to further improve and clarify several important aspects of the manuscript.
- Our detailed responses are provided below and are structured directly under each individual comment.

**General Comments**

**Relevance Beyond Machine Learning**

While the dataset is positioned primarily as a resource for machine learning (ML) approaches, its value extends well beyond this scope. The authors could broaden the framing of the paper to highlight its usefulness for a wider range of hydrological and environmental applications, including traditional modeling approaches, process understanding and decision-support tools.

We thank the reviewer for this valuable comment. While the dataset is indeed designed as a
benchmark for machine learning applications, we agree that its scope and value extend well
beyond ML. Following the suggestion, we have expanded the Introduction (Lines 78–80) to
emphasize additional applications, including groundwater trend analysis, drought and climate
impact assessment, and the calibration and validation of traditional hydrological models.

**Discussion of Uncertainties**

Although the authors have clearly put effort into filtering implausible values and improving data quality, the manuscript does not adequately address uncertainties. In particular: i) Measurement uncertainties in groundwater levels are not discussed and ii) Uncertainties in derived or static attributes (e.g., recharge estimates, soil data) are likely more impactful for modeling and deserve attention. A discussion of these uncertainties would help readers better understand the dataset's limitations and appropriate use.

• Thank you for that comment. We fully agree that a discussion of uncertainties is essential to clarify the dataset's limitations and appropriate use. Following the suggestion, we have added a new subsection "Uncertainties and limitations" in the Data and Preprocessing section. This subsection explicitly discusses (i) uncertainties in groundwater level measurements (e.g., sensor drift, manual reading errors, inconsistencies in reference datums, and resampling artifacts from heterogeneous measurement intervals) and (ii) uncertainties in

environmental and meteorological attributes, including scale and resolution differences, simplifications in categorical classifications, temporal inconsistencies in multi-year datasets, and uncertainties inherited from model-based products such as groundwater recharge or reanalysis data.

**Interpolated vs. Raw Data**

The manuscript mentions interpolation of missing values in groundwater time series. While the interpolation is reasonable, every method carries assumptions and trade-offs. I recommend that both versions—interpolated and raw (with gaps)—be made available. This would increase transparency and provide flexibility for future users.

• This functionality is already included in the dataset. Each groundwater level entry is accompanied by a binary flag (*GWL\_flag*) indicating whether the value was directly observed (*True*) or imputed (*False*). This allows users to either work with the fully gap-filled dataset or restrict their analyses to raw observations only. To avoid misunderstandings, we have clarified this more explicitly by adding the information not only in Section 3.1 (*Dynamic time series data*) but also in Section 2.2.5 (*Data Imputation*).

The interpretation of NSE values is debatable. Stating that NSE  $\approx$  0.5 is "relatively strong" is not convincing in this context, as groundwater levels—compared to surface water flows—are typically smoother and have longer response times and are typically easier to predict. NSE values <0.5, especially at a weekly resolution, suggest that model performance is modest and should be described as such.

- We acknowledge the reviewer's concern and have revised our interpretation to be more
  conservative. In the revised manuscript, we now describe NSE ≈ 0.5 as "acceptable" rather
  than "relatively strong" and have removed the direct comparison with surface water
  modeling benchmarks.
- However, we respectfully disagree with the assertion that groundwater levels are inherently easier to predict than surface water flows. While groundwater time series may appear smoother due to subsurface buffering effects, several factors make groundwater prediction particularly challenging: (1) Subsurface heterogeneity: Aquifer properties (hydraulic conductivity, storage coefficients) vary spatially over orders of magnitude and are difficult to characterize comprehensively; (2) Diffuse and delayed processes: Recharge occurs through complex vadose zone processes with highly variable time lags that are difficult to quantify; (3) Unobserved anthropogenic influences: Pumping rates, irrigation return flows, and land-use changes significantly affect local groundwater dynamics but are rarely monitored or available as model inputs; (4) Multi-scale interactions: Groundwater systems integrate processes across multiple temporal and spatial scales, from local heterogeneities to regional flow patterns.

These complexities lead to much more different types of dynamics, from quasi inert over rather smooth and seasonality-dominated to short-term changing and rather flashy dynamic types. This limits model performance, especially of global models, which in groundwater show no consistent accuracy gains over singlewell models, depend strongly on dynamic similarity rather than dataset size, and generalize well only to similar wells. (https://doi.org/10.5194/egusphere-2025-4055).

In this context, (for example Line 264) the benchmark modeling is intended to provide transparency rather than achieve optimal predictive performance. However, the interpretation of model performance (e.g., identifying locations where other drivers may be important) seems somewhat optimistic. For instance, Figure 5 demonstrates that even at sites without apparent additional drivers, machine learning performance can still be limited. A more nuanced discussion of these limitations would strengthen the paper. Moreover, when the model performance is not fully mature, one might question the rationale for conducting this analysis, as it could potentially create confusion—particularly when the boundary between poor model performance and the influence of external drivers becomes blurred.

- We appreciate this comment and agree that the limitations of model interpretation need to be stated more explicitly. We would like to clarify, however, that the notion of "apparent additional drivers" is problematic in the groundwater context. In most cases, the relevant external influences are not directly apparent in the time series (with the possible exception of surface water interactions). Instead, many drivers such as groundwater abstraction, artificial recharge, or construction dewatering are largely hidden, rarely monitored, and therefore not represented in the dataset. Similarly, we find the idea of a "boundary" between poor model performance and the influence of external drivers difficult to define. In practice, poor performance is often precisely the result of unobserved external drivers, and thus the distinction is not a sharp boundary but rather a continuum.
- We also note that the reviewer refers to Figure 5 (NSE distribution as boxplots and CDF),
  where such drivers cannot be identified. We assume that this comment may instead relate to
  Figure 6 (spatial distribution of NSE values), where regional patterns and local anomalies are
  more visible.
- To avoid misunderstanding, we have revised the manuscript to provide a more cautious interpretation of the benchmark results and to explicitly emphasize these limitations. Specifically, we now state that low NSE values cannot be unambiguously attributed to external drivers, as limited performance may equally stem from remaining data uncertainties, the restricted input feature set, or the simplicity of the benchmark architectures. We further highlight that these aspects are intertwined, and that the distinction between unobserved external influences and internal model limitations should be seen as a continuum rather than a sharp boundary (Lines 341-345).

Additionally, it is not clear to me whether the NSE is based only on observed measurements or if it also includes interpolated values. The comparison should only be made using the actual measurements, not values that are already derived from a method or model.

- We appreciate the suggestion. NSE is computed on the full, flagged series; imputed values are rare in the test period (mean 0.9%, max 3.8% per well). Consistently, the station-wise differences between All and True-only NSE are negligible (mean absolute ≤ 0.003; maxima ≤ 0.085; Table 1). Reporting a second set of observed-only scores would therefore add unnecessary complexity without changing any conclusions. For transparency, we state explicitly that imputed values (GWL\_flag) are included, and users can easily reproduce True-only metrics if desired.
- **Table 1.** Station-wise NSE comparison of three models on the GEMS-GER dataset, reported as **All** / *True-only* (All = with imputations; *True-only* = measured data only, italic). Columns: NSEmin, NSEmean, NSEmed, NSEmax; **Diffmax** = max station-wise (*True-only All*), **Diffabs.mean** = mean absolute station-wise difference.

| Model   | NSE min | NSE mean | NSE med   | NSE max | Diffmax | Diff abs.mean |
|---------|--------------------|---------------------|----------------------|--------------------|---------------------------|--------------------------|
| dynstat | -4.314 / -4.314    | 0.395 / 0.395       | 0.496 / 0.496 | 0.905 / 0.907      | 0.079                     | 0.0024                   |

| dynonly | -6.781 / -6.781        | 0.316 / 0.315 | 0.316 / 0.315        | 0.884 / 0.886 | 0.085 | 0.0026 |
|---------|------------------------|---------------|----------------------|---------------|-------|--------|
| single  | -3.466 / -3.466 | 0.408 / 0.407 | 0.517 / 0.517 | 0.938 / 0.938 | 0.040 | 0.0027 |

Furthermore, I am wondering why no values are provided at least for the major rivers. Especially when using machine learning approaches, it is crucial to integrate the relevant processes—such as groundwater or surface water interactions—otherwise a good fit may be achieved for the wrong reasons. In my opinion, this should definitely be done, particularly because river data are generally reliable and easily accessible.

- We fully agree that groundwater—surface water interactions can be important drivers of groundwater level dynamics, particularly close to rivers. However, we decided not to include river discharge or gauge data in the present dataset for several reasons.
- Limited applicability: Only a small fraction of the >3,000 wells are located close enough to
  major rivers for such data to be directly relevant. Including river attributes for just this subset
  would compromise the consistency and comparability of the dataset. Extending it
  systematically to all rivers, however, would require substantial methodological choices (e.g.,
  definition of distance thresholds, assignment to upstream/downstream gauges) and is
  beyond the scope of this work.
- Availability in other datasets: Reliable river discharge data for Germany are already available
  through the CAMELS-DE dataset (<a href="https://doi.org/10.5194/essd-16-5625-2024">https://doi.org/10.5194/essd-16-5625-2024</a>). Users who
  wish to explicitly study groundwater—surface water interactions can readily combine GEMSGER with CAMELS-DE or other hydrological data products, avoiding duplication of existing
  resources.
- **Mutual interactions:** While river stages may influence nearby groundwater levels, the reverse is also true, as baseflow contributions constitute a substantial fraction of river discharge (on average ~59 % globally according to recent estimates). This two-way coupling highlights the conceptual complexity of integrating surface water and groundwater datasets at national scale, which would require careful design to ensure consistency.
- For these reasons, and to maintain the focus on a standardized benchmark dataset, we
  deliberately excluded river data in this first release. Nevertheless, we acknowledge the
  relevance of this aspect and see the integration of river attributes as a valuable extension for
  future studies, particularly in regional analyses where suitable well—river assignments can be
  defined more precisely.

**Specific Comments**

Line 24: The statement that ML approaches "can/should be used for assessing the impact of climate change" is too strong. ML models trained on historical data may not extrapolate reliably to unprecedented climate conditions (e.g., prolonged droughts, multi-year droughts which may create a system trigger point). The statement should be rephrased to reflect these limitations.

• We thank the reviewer for this important remark. In the revised manuscript, we have removed the original strong statement on the role of ML for assessing climate change impacts. Instead, we now emphasize the limitations more cautiously. Specifically, we added a note that "future climate variability and change may further complicate groundwater projections, underlining the need for robust and transparent data foundations." This phrasing acknowledges the reviewer's concern and ensures a more balanced framing.

Line 31: There is a modeling approach between ML and fully physically based models—simplified point-scale or lumped-parameter models (e.g., AquiMod, Pastas). As noted by Bakker & Schaars

(2019; 10.1111/gwat.12927), these models can provide efficient and accurate groundwater forecasts with lower data requirements. Including this perspective would improve the model concepts completeness although the approach is not used here (https://doi.org/10.1016/j.envsoft.2014.06.003; https://doi.org/10.1016/j.jhydrol.2023.130120; https://doi.org/10.1111/gwat.12819 among other references) .

Thanks for raising this important point. We have revised the introduction to acknowledge intermediate modeling approaches. Specifically, we now discuss lumped-conceptual models such as AquiMod (Mackay et al., 2014) and Transfer Function—Noise (TFN) models implemented in Pastas (Bakker et al., 2019; Zaadnoordijk et al., 2019; Collenteur et al., 2023).
 These approaches can indeed provide efficient and accurate groundwater forecasts with relatively low data requirements and are widely used in practice. However, since they typically rely on site-specific parameter calibration, their applicability across regions and diverse hydrogeological settings may be constrained. We now highlight this aspect in the manuscript as a motivation for the complementary exploration of more generalizable machine learning methods.

Line 34: The statement "...even when observational data are limited" is oversimplified. It should clarify that transfer learning or cross-site modeling may help in data-scarce regions, but the limitations of sparse observations must be acknowledged.

We thank the reviewer for this valuable remark. We agree that our original statement was
too strong and have revised it for clarity. The text now emphasizes that, although ML can
capture complex non-linear relationships, sparse observations remain a limitation. We also
highlight that recent advances such as transfer learning and cross-site modeling may help
extend ML applications to data-scarce regions, but that these approaches cannot fully
compensate for a lack of observational data.

Figure 2: Please clarify why no data remain for HB after filtering. Unlike SL or HH, where the absence of data is understandable, it appears that HB has usable data that could potentially be gap-filled. More details and explainations would help.

• The absence of wells from Bremen (HB) after filtering is indeed due to the same criteria applied to all states (≤20 % missing data and ≤12-week gap within 1991–2022). Thus, no monitoring wells from HB, HH, or SL fulfilled the requirements and remain in the final dataset. While this is visually more evident for HH and SL in Figure 2, we acknowledge that it is less obvious for HB. To clarify, we have revised the figure caption accordingly.

Figure 4: This figure illustrates the type of uncertainty that deserves more attention (as mentioned above). For example, recharge estimates often vary widely depending on methodology (see for example https://hess.copernicus.org/articles/25/787/2021/, https://doi.org/10.1029/2022GL099010 although the scale is different), and this variability may affect ML performance. I am not proposing to do the modelling with multiple recharge or precipitation products (or any other variable) because this is just unrealistic but discussing these uncertainties explicitly would be helpful.

 We have added a new section, "Uncertainties and limitations," providing a concise, crosscutting discussion of uncertainty across measurements, environmental/meteorological attributes, and model-derived variables. The text explains that uncertainties arise from station-network representativeness, scale and resolution mismatches, categorical generalization, temporal inconsistencies, and structural assumptions in gridded and reanalysis products. These uncertainties propagate through data processing, affecting well-level accuracy and transferability. For detailed uncertainty quantifications beyond this paper's scope, readers are referred to the original data sources cited in the tables.

---

## Author Comment (AC2)

The manuscript describes a important groundwater level dataset for benchmarking machine learning models. I like the dataset because it is both important and timely for multiple disciplines, including hydrology, hydrogeology, and data science, etc. There is no doubt that it will be widely used for machine learning development, similar to the CAMELS datasets. I also like the manuscript because it is clear and concise. The dataset is well organized and follows the FAIR principles. I have few comments on the manuscript and dataset for the authors' consideration when they do some minor revisions.

- We would like to sincerely thank Reviewer 2 for the very positive and constructive feedback on our manuscript essd-2025-321. We greatly appreciate the reviewer's recognition of the dataset's scientific relevance, clarity, and adherence to FAIR principles.
- Our detailed responses are provided below and are organized directly under each individual comment for clarity.

**Major comments.**

Are the groundwater levels from shallow wells, deep wells, or a combination of both? Please clarify this information in the manuscript.

We thank the reviewer for this helpful remark. The dataset indeed comprises a combination of shallow and deep observation wells across Germany, covering both unconfined and confined aquifer conditions. Corresponding information is included in the metadata (fields *Depth*, *UpFilter*, *LoFilter*, *ScrLength*, *PreState*) and listed in Table 2. We have clarified this in the revised manuscript (Section 3.3, "Site-specific static data") by adding the following sentence:

"The resulting dataset covers both shallow and deep monitoring wells under unconfined and confined conditions, as indicated by the available depth and pressure state metadata."

Although the imputation rate in the dataset is quite low, would it be good to include a synthetic test to evaluate the quality of the imputation (i.e., how well or poorly it performs)?

- We appreciate this thoughtful suggestion. The overall imputation rate in the dataset is indeed very low (mean 0.9 %, maximum 3.8 % during the test period), and missing values are typically short and evenly distributed. For this reason, we did not perform a separate synthetic test, as the influence of imputation on model performance was expected to be negligible. To verify this assumption, we compared benchmark model results based on the imputed dataset with an otherwise identical version where missing values were left unfilled. The resulting differences in RMSE, R², and Bias were marginal (see our response to Reviewer 1, Tab 1), confirming that the imputation had no relevant impact on predictive performance.
- Nevertheless, the imputation procedure was carefully designed to minimize potential bias: it
  leverages correlated neighboring wells as predictors, applies a block-wise strategy with
  temporal overlap to preserve seasonal dynamics, and explicitly flags all imputed values using
  the binary variable GWL\_flag. This enables users to independently assess, validate, or reimplement the imputation process if desired.

The role of static covariates in machine learning model development (e.g., CNN or LSTM) is somewhat limited if data from all 3,207 wells are used for training, validation, and testing. I suggest the authors perform a spatiotemporal testing experiment — for example, hold out a group of wells entirely from the training set and use them only for testing — to evaluate whether the static covariates improve model performance.

- We appreciate this valuable suggestion and fully agree that spatial hold-out validation represents an important next step for assessing the contribution of static covariates to model generalization. However, the present study aims to establish the dataset and provide initial benchmark models that demonstrate its usability rather than to exhaustively explore all possible validation strategies.
- Our focus was therefore on transparent and reproducible baseline models using a consistent temporal split across all wells. Spatially independent evaluation experiments (such as leaveone-region-out or stratigraphic or clustered hold-out tests) are indeed planned for follow-up studies using the GEMS-GER dataset and are explicitly encouraged within its documentation.
- To reflect this point, we have added a short statement in the Conclusions section emphasizing that future work may include spatiotemporal validation experiments to systematically evaluate the role of static features in model transferability.

**Minor comments.**

Lines 91–101: Consider presenting the names in a table and including the number of wells associated with each. This would make it easier for readers to understand.

• We appreciate this helpful suggestion. The distribution of wells across the 16 German federal states is already visualized in Figure 2, which shows the number of wells per state at different preprocessing stages. For clarity, we slightly revised the text in Section 2.1 to explicitly refer to this figure when listing the data-providing authorities.

Line 111: The acronym LANUV has already been introduced earlier; the full name is not needed here.

• We thank the reviewer for pointing this out. We have removed the repeated full name of LANUV and now refer to the authority only by its acronym in this section.

Section 2.2.4: Please add one or two sentences explaining the possible reasons for the identified outliers.

We thank the reviewer for this helpful suggestion. We have added two sentences at the end
of Section 2.2.4 explaining that most outliers were related to technical or anthropogenic
factors rather than natural groundwater dynamics (e.g., sensor malfunction or recalibration,
data logger replacement, or short-term impacts from construction, pumping, or maintenance
activities near the observation wells).

Line 186. You mean 'density' here, is that calculated by number of wells divided by area? or MHD1 has the highest number of wells?

• Yes, "density" refers to the number of monitoring wells per unit area (wells / km²) within each Major Hydrogeological District (MHD). We have clarified this in the text accordingly.

Lines 190–200: Are these variables listed in Table 2? If so, note that they could also serve as static covariates for machine learning model development.

We deliberately did not include these dynamic indicators as additional static covariates in the
dataset to avoid further increasing the already extensive set of static attributes (> 50 features).
 Moreover, they can easily be recomputed from the provided groundwater level and
meteorological time series, allowing users to decide individually whether and how to integrate
them into their own feature-engineering workflows.

 While such derived variables could potentially be used as static or quasi-static covariates in dedicated model experiments, the benchmark models presented here were intended as a compact and transparent "add-on" to illustrate the dataset's usability rather than an exhaustive modeling exercise. We therefore intentionally kept the setup simple and encourage future studies to explore extended feature combinations.

Section 3: While the cleaned and quality-controlled groundwater-level time series are very useful, have the authors considered including the original (raw) dataset as well?

• We decided not to include the unprocessed raw data in the public release for several reasons. First, the raw groundwater-level series contain technical artifacts and implausible spikes that could easily lead to misinterpretations if used without the accompanying quality-control procedures. Second, the published dataset provides full transparency through the GWL\_flag variable, which allows users to identify and exclude imputed values, effectively reconstructing the validated observations. Moreover, maintaining a second, unfiltered data version would create redundancy and complicate version control, potentially undermining reproducibility. Finally, the main purpose of GEMS-GER is to serve as a harmonized, quality-assured benchmark dataset rather than a raw data archive, in line with ESSD's emphasis on usability and scientific robustness